# Photoplethysmograph Based Biofeedback for Stress Reduction under Real-Life Conditions in Healthcare Frontline

Emese Rudics [1,2], Ádám Nagy [1,*], József Dombi [3], Emőke Adrienn Hompoth [1], Zoltán Szabó [1], Rózsa Horváth [4], Mária Balogh [4], András Lovas [5], Vilmos Bilicki [1] and István Szendi [1,6]

1 Department of Software Engineering, University of Szeged, Dugonics Square 13, 6720 Szeged, Hungary
2 Doctoral School of Interdisciplinary Medicine, Department of Medical Genetics, University of Szeged, Somogyi Béla Street 4, 6720 Szeged, Hungary
3 Department of Computer Algorithms and Artificial Intelligence, University of Szeged, Árpád Square 2, 6720 Szeged, Hungary
4 Emergency Department, Kiskunhalas Semmelweis Hospital, Dr. Monszpart László Street 1, 6400 Kiskunhalas, Hungary
5 Department of Intensive Care Unit, Kiskunhalas Semmelweis Hospital, Dr. Monszpart László Street 1, 6400 Kiskunhalas, Hungary
6 Department of Psychiatry, Kiskunhalas Semmelweis Hospital, Dr. Monszpart László Street 1, 6400 Kiskunhalas, Hungary
* Correspondence: adam.nagy@inclouded.hu

**Abstract:** Biofeedback (BF) therapy methods have evolved considerably in recent years. The best known is biofeedback training based on heart rate variability (HRV), which is used to treat asthma, depression, stress, and anxiety, among other conditions, by synchronizing the rhythm of breathing and heartbeat. The aim of our research was to develop a methodology and test its applicability using photoplethysmographs and smartphones to conduct biofeedback sessions for frontline healthcare workers under their everyday stressful conditions. Our hypothesis is that such a methodology is not only comparable to traditional training itself, but can make regular sessions increasingly effective in reducing real-life stress by providing appropriate feedback to the subject. The sample consisted 28 participants. Our proprietary method based on HRV biofeedback is able to determine the resonance frequency of the subjects, i.e., the number at which the pulse and respiration are in sync. Our research app then uses visual feedback to help the subject reach this frequency, which, if maintained, can significantly reduce stress. By comparing BF with Free relaxation, we conclude that BF does not lose effectiveness over time and repetitions, but increases it. This paper is our pilot study in which we discuss the method used to select participants, the development and operation of the protocol and algorithm, and present and analyze the results obtained. The showcased results demonstrate our hypothesis that purely IT-based relaxation techniques can effectively compete with spontaneous relaxation through biofeedback. This provides a basis for further investigation and development of the methodology and its widespread use to effectively reduce workplace stress.

**Keywords:** biofeedback; psychiatry; healthcare; work-related stress; guided relaxation

## 1. Introduction

Biofeedback therapy is a non-invasive method of stress management that enables the restoration and maintenance of the vegetative balance through precise measurement and real-time feedback of the current state of physiological processes. There are different types of biofeedback therapies depending on which physiological process is being monitored and controlled during the treatment. For example, physiological processes such as respiration rate, skin temperature, heart rate, blood pressure, electrodermal response, muscle activity and brain activity can be used during biofeedback training to learn and develop self-regulation [1]. The biofeedback loop consists of a biosensing unit, which registers the unconscious physiological process, and transmits it through the data transfer

unit to the data processing unit, where the data is analyzed. Then the analyzed data is presented on the feedback unit and the subject regulates his or her physiological process according to the audiovisual feedback [2]. During biofeedback therapy subjects synchronize their physiological processes with the feedback stimuli, inducing mental and physical changes [1].

Heart rate variability (HRV) biofeedback training is based on the monitoring of heart rate variability [1]. HRV biofeedback training is used in the treatment of chronic diseases. It improves asthma conditions, depression, sleep disturbances, hypertension, coronary artery disease, inflammatory conditions, and fibromyalgia, among others [3]. The training can also be an effective way to reduce stress and anxiety symptoms [4]. Reducing stress level works by synchronizing breathing patterns with heart rate. According to the HRV biofeedback training protocol, the first step is to measure the unique resonant frequency and learn abdominal breathing. In abdominal breathing with lips pressed together, the subject inhales through the nose into the stomach, and exhales through the mouth slowly to avoid hyperventilation [5]. The HRV biofeedback training always includes visual feedback of the heart rate, usually a heart rate graph, to help participants adjust their breathing to their heart rate (inhaling as heart rate increases, exhaling as heart rate decreases). During HRV biofeedback training sessions with abdominal breathing, a unique resonance frequency is learned [5,6].

According to the HRV biofeedback training protocol the method becomes efficient after five training sessions and regular 20 min home practices. The protocol is used in both clinical practice and research [5]. Other studies have found that HRV biofeedback training can be efficient after practicing for three to four weeks [7], furthermore, even a single session of HRV biofeedback training can increase the subjective feeling of relaxation [8].

A comparison between some of the most notable relaxation trainings and methodologies can be seen in Table 1.

HRV biofeedback training reduces both laboratory-induced [9] and real-life stress and anxiety [10] and can also be used in everyday stress management as well [2]. Compared to other effective stress reduction methods, such as physical activity and mindfulness meditation, HRV biofeedback training is almost as effective in reducing the perceived stress and anxiety and improving psychological well-being [11]. According to Lin and colleagues [8] there is no significant difference in the intensity of subjective relaxation induced by a single session of HRV biofeedback training or autogenic training. In addition, HRV biofeedback training has been found to be a useful method for reducing subclinical anxiety symptoms [12]. Chung and colleagues [12] used a wearable sensor during their eight-week HRV biofeedback training to monitor heart rate variability and provide feedback during HRV training. In addition, a smartphone application was connected to the sensor to provide visual feedback to the subject about breathing patterns. The eight-week HRV biofeedback training significantly reduced subclinical symptoms of anxiety and depression [12]. Stress and anxiety decreased after HRV biofeedback training in both male athletes [10], and soldiers [9].

Our goal was to develop a procedure to help frontline healthcare workers in real-life conditions to reduce real-world workplace stress while working at an increased intensity and fluctuating pace. Working in the healthcare system can be extremely stressful [13–15]. Healthcare professionals who work in Intensive Care Unit (ICU) or Emergency Department (ED) may experience emotional distress on a daily basis [14,15], which could increase the risk of burnout and compassion fatigue [14].

The aim of our study is to validate the efficiency of the HRV biofeedback training compared to relaxation exercises under real life conditions, when the trainer is not available, and training is only feasible via IT technology. Another goal of our research is to investigate whether HRV biofeedback application results in increased effectiveness in stress reduction with repetition compared to regular relaxation exercises.

**Table 1.** Comparison of major relaxation methodologies.

| | Groups | Tool | Personal Guidance | Sessions, Exercises | Elapsed Time | Result |
|---|---|---|---|---|---|---|
| Lin et al. (2020) [8] | - HRV-BF<br>- AT group | - Application vs.<br>- Autogenic training | - None vs.<br>- Yes | 1 training session in both cases (15 min) | (1 session) | Significant time main effect on relaxation scores, no significant difference between methods |
| Dziembowska et al. (2016) [10] | - Biofeedback<br>- Control | emWave biofeedback tool | - For one training session<br>- None | - 1 training and 10 practice sessions<br>- Only the 10 practice sessions | 3 weeks | - BF group: significant decrease in state anxiety<br>- No decrease in state anxiety |
| Van Der Zwan et al. (2015) [11] | - PA<br>- MM<br>- HRV-BF | - PA<br>- MM<br>- StressEraser | Yes, 2 h introduction meeting | Daily exercises, on the first week 10 min, 2nd week 15 min, 3–5th weeks 20 min daily | 5 weeks | Stress related symptoms decreased over time |
| Chung et al. (2021) [12] | HRV-BF group | Lief Smart Patch and phone application | Once during a phone call | Three guided breathing exercises each day for 40 out of 56 days, each exercise lasts for 3 min | 8 weeks | Reduced anxiety symptoms |

Glossary: HRV-BF (Heart rate variability biofeedback); AT (autogenic training), PA (Physical activity), MM (Mindfulness meditation).

## 2. Materials and Methods

### 2.1. Participants

The sample size was determined by preliminary, approximate power analysis using the R "pwr" function library based on Goessl et al. [4] meta-analysis. In the two-sample *t*-test power calculation, the effect size was set at 0.85 and the significance level at 0.05, resulting in N = 29.05. Due to attrition and other data loss, we planned to include 36 people in the study.

Thirty-three participants were recruited for the study, but five participants were excluded because of improper use of the HRV biofeedback application. The final sample of the study consisted of twenty-eight healthcare professionals (23 women, 5 men, mean age M = 44.25, standard deviation of age SD = 9.95 ) working in the Intensive Care Unit (ICU), Emergency Department (ED) and Psychiatry Unit (PU) of Semmelweis Hospital in Kiskunhalas, the Teaching Hospital of the University of Szeged. Ten of the participants were healthcare professionals in ICU (8 women, 2 men, mean age MICU = 42.20, standard deviation of age SDICU = 11.71), thirteen participants worked in ED (11 women, 2 men, mean age MED = 45.38, standard deviation of age SDED = 10.57) and five in psychiatry (4 women, 1 men, mean age MP = 45.40, standard deviation of age SDP = 2.88 ). The study sample consisted of three physicians, one resident, nineteen nurses, and five medical records specialists. Seven participants had secondary school, eleven participants had graduation and 10 participants had higher educational level. Twenty-nine percent of the sample had 10 years or less of medical work experience. According to marital status eleven participants (39%) were married, eight (29%) were unmarried and nine participants (32%) were in a relationship. Inclusion criteria were age between 18 and 65 years and eligibility for work in ICU, ED or PU. Exclusion criteria were the presence of a primary or secondary mental disorder and/or withdrawal of consent for the examination. Prior to the examination participants were informed about the research and signed an informed consent. The study was approved by the National Public Health Center (6860-12/2022/EUIG). Participants received financial compensation for their participation in the study.

*2.2. Methods*

2.2.1. Heart Rate Variability Biofeedback Application

The biofeedback application was developed to test the stress-relieving efficiency of HRV biofeedback method. The application was downloaded onto tablets and pulse oximeters were connected to the tablets to continuously monitor the participant's heart rate during the heart rate variability biofeedback training. The application starts with a login page, where participants can register and later sign in. After signing in for the first time, the biofeedback application recommends measuring the participant's unique resonance frequency. The resonance frequency interface consists of a continuously pulsating circle that helps participants maintain a steady breathing rate (top left corner of the interface). The measured frequencies are 16, 14, 12, 10, 9, 8, 7.5, 7, 6.5, 6, 5.5 and 5 breaths per minute. The pulse curve of the participant during the measurement can be seen at the bottom of the user interface. Information about the participant's current resonance frequency, the elapsed time since the beginning of a measurement, the number of breaths, and the start-stop button are located at the top right corner of the interface. After the biofeedback application calculates the unique resonance frequency, the time of the measurement and the number of the resonance frequency appear at the top of the interface. After calculating the resonance frequency, the experimental measurement (HRV biofeedback training—HRV condition) becomes accessible.

The experimental measurement consists of six phases, with the application (simply titled as Biofeedback) providing instructions in all of them. In the first and fifth phases, the Biofeedback application instructs the participants to move freely in the examination room for one minute, the expiration of the time is signaled by a tone. In the second and fourth phases, before and after the HRV BF training or control condition (free relaxation), the application displays the STAIS-5 questionnaire [16] and the Distress Thermometer [17] to measure the level of perceived stress. The third phase includes the HRV biofeedback training condition or the control condition (free relaxation). The Biofeedback application alternates between the two conditions. The control condition includes the free relaxation in which participants spend their time freely (using mobile phone, reading news on phone, messaging, or relaxing). The HRV and the control condition last for six minutes, and the expiration of the provided time is signaled by sound. Finally, the sixth step is to save the measurement.

The interface of the HRV biofeedback training condition is similar to the interface of resonance frequency measurement. The training interface is divided into three areas. At the top of the screen is information about the participant's last unique resonance frequency and the time of measurement, below that is visual feedback about the user's performance. In the center of the screen, in the left corner the visual feedback on breath control (a continuously pulsating circle) is placed, in the right corner, the elapsed and remaining time until the end of the measurement (maximum 360 s), the number of breathings and start-stop button is located. Instructions about abdominal breathing are located at the bottom of the screen. A save bottom is also placed here after the training ends.

2.2.2. Psychological Questionnaires

The Biofeedback application contains the shortened Hungarian version of the Spielberg State-Trait Anxiety Inventory (STAIS-5) to assess the level of perceived distress [16]. The questionnaire contains five items that can be evaluated on a four point Likert scale (from "not at all" to "very much so") [16]. The Distress Thermometer (DT) [17] is used to assess the level of distress. DT is a visual analogue scale consisting of one item that can be rated on a scale of one to ten from "no distress" to "extreme distress" [17].

2.2.3. Clinical and Demographic Interview

Mini International Neuropsychiatric Interview (M.I.N.I.) [18] is a structured diagnostic interview used to identify potential psychiatric disorders. The 15-min interview screens for psychiatric disorders defined by ICD-10 and DSM-IV [18]. A demographic questionnaire

was also completed, collecting data about participant's age, gender, relationship status, educational level, profession, hospital ward in which the participant worked at the time of the research and the number of years worked in that hospital ward, the number of years worked in the hospital up to the day of examination, mental and somatic disorders, medications taken during the examination, and mental disorders in first-degree relatives.

2.2.4. Tools Used in the Research, but Not Analyzed

The shortened Hungarian version of Temperament and Character Inventory [19], the Hungarian version of the Temperament. Evaluation of Memphis, Pisa, Paris and San Diego (TEMPS-A) [20], theHungarian version of the Behaviour Inhibition System/Behaviour Activation System Scales (bis-bas scales) [21], MOX3 ECG and Physical Activity recording system, physical diary, MOX Activation application.

*2.3. Procedure*

Prior to the intervention, participants attended a personal counseling session. The personal consultation began with the participant registering in the Biofeedback application with a previously established unique identifier and a freely chosen password,. Participants were then taught diaphragmatic breathing and the use of the Biofeedback application, then the unique resonance frequency was measured with the application. Then, the use of the MOX3 ECG and Physical Activity recording system (Maastricht Instruments BV) was demonstrated, the device was handed out to the participants and the MOX Activation mobile application (which was also made by us) was downloaded onto the participants' smartphones. Once the MOX Activation application was installed, the participant's signed in with the same identifier and password used in the Biofeedback application, and the research assistant demonstrated how to use the application. Participants were also given a physical diary for recording their daily activities during the experiment. At the end of the consultation, demographic and psychological questionnaires were completed and the Mini International Neuropsychiatric Interview was conducted. The personal consultation took 30 min. After the personal consultation, participants wore the MOX3 ECG and Physical Activity recording system continuously for five days, while using the MOX Activation application and recording their daily activity in the physical diary. The Biofeedback application was used in parallel with the MOX3 device. Participants used the Biofeedback application on work days during breaks in a quiet room on each hospital ward. Use of the Biofeedback application took approximately 10 min.

*2.4. Algorithm*

For the algorithm to work properly, we must first calculate the resonance frequency for each new user, which will serve as a basis for subsequent evaluations. Therefore, the first step for each patient is to perform the breathing exercises. This involves recording 1 min of photoplethysmogram (PPG) data while the patient breathes at the specified frequency (16, 14, 12, 10, 9, 8, 7.5, 7, 6.5, 6, 5.5 and 5 breaths per minute in that order). The first 15 s are ignored in these measurements because interfering, distorting signals are too common there. The PPG data can be used to clearly identify heartbeats and plot the Heart Rate (HR) curve. At the end of the minute, the HR curve is median filtered to remove distortions caused, for example, by small hand movements, and then compared with the respiration curve. This is done by calculating the correlation between the two sets of values. The mean values are subtracted from both curves, and then the values of the respiration curve are taken only at the times when a heartbeat occurred. These values are multiplied together and the multiplications are added together and then multiplied by Heart Rate Variability (HRV) to increase accuracy. The respiratory rate that has the highest value is marked as the subject's resonance frequency.

Once the resonance frequency for the particular patient is known, it is possible to perform the biofeedback breathing exercises with the application. During these exercises, the application first collects PPG data with the sensor for 60 s. The mean values are

removed from the collected data, which is then interpolated to a uniform distribution and then subjected to the Fast Fourier Transform (FFT) which provides the frequency and amplitude resolution of the collected plethysmogram signal.

Following the initial evaluations, the methodology was developed so that the frequency associated with the maximum amplitude was considered as an estimate of the respiration frequency and the first two frequencies were ignored because they did not belong to the respiration in any of the analyzed cases. In the case of two frequencies with the same maximum amplitude, the higher frequency was used, since it was always closer to reality in the cases studied.

The estimate of respiration is then refined by autocorrelation. The resonance frequency is the frequency at which the respiration and pulse frequencies are closest to each other.

After the first 60 s, recalculations follow in a 10 s sliding window: after 10 s, the application automatically recalculates the last 60 s of data until it reaches or exceeds the frequency previously specified as the resonance frequency.

### 2.5. Data Processing

During the data cleaning, we deleted the datasets where participants performed the two different types of relaxation directly after each other. The datasets in which the STAIS-5 or the DT scores were minimal, indicating that no stress was present at baseline, were also deleted. In the end, we filtered out the 5 (out of 33) subjects mentioned above, leaving 115 of the 233 records. The STAIS-5 has a discrete scale of 1 to 4 points, whereas DT has a discrete scale of 1 to 10 points. We rescaled these to the interval of 0–1 then shifted them to the 1–2 to be able to calculate the relative difference between the "before" and "after" scores. The relative difference was determined as follows: We subtracted the "before" scores from the "after" scores and then divided by the "before" scores. We created three variables:

- thermo_diff—relative difference in the DT
- stai_diff—relative difference in the mean of the STAIS-5
- diff_all—relative difference in the mean of the STAIS-5 and the DT

These variables and their behavior on either HRV conditioning or free relaxation can be seen in Figure 1. During this step we did not delete any outliers. These tree variables are slightly continuous, and cannot be described by a single component distribution, so we had to use nonparametric statistical tests for the analysis.

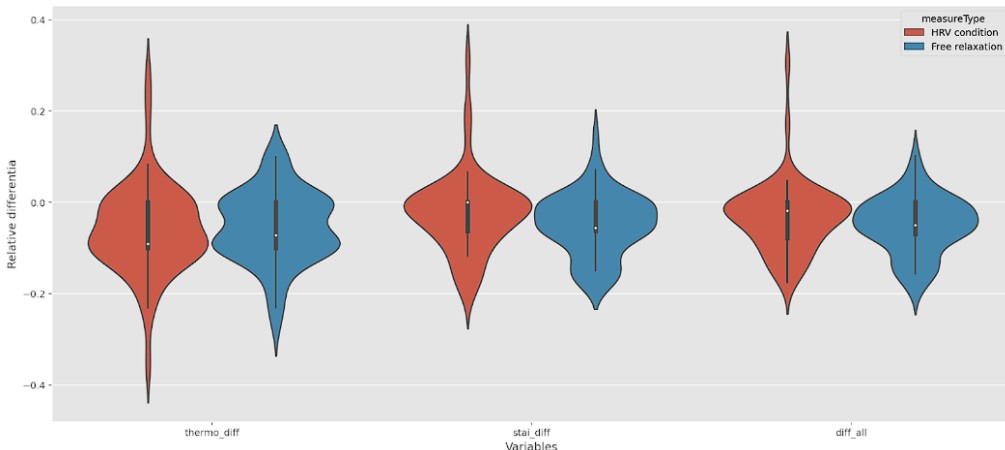

**Figure 1.** The violin plot of three variables creating during data processing and their relative differences on different relaxation methodologies.

Table 2 shows the means and standard deviations of the variables in each group. The negative values represent the reduction of stress and other parameters. As we can see, only the DT parameter (thermo_diff) obtained better results in the HRV condition group.

**Table 2.** Mean and standard deviation of the three variables on HRV and Free relaxation methodologies.

| Variable | HRV Condition Mean (Std) | Free Relaxation Mean (Std) |
|---|---|---|
| thermo_diff | −0.061 (0.096) | −0.055 (0.078) |
| stai_diff | −0.024 (0.086) | −0.048 (0.068) |
| diff_all | −0.033 (0.077) | −0.05 (0.062) |

*2.6. Statistical Analysis*

2.6.1. Non-Parametric Tests

To test the difference between the two methods (HRV condition and Free relaxation) we used the nonparametric Mann-Whitney U test. The null hypothesis of this test is that the samples are from the same population. We tested the efficiency of the two methods using the Wilcoxon signed rank test. The test uses the location of a population based on a sample of data. In our case the population came from the relative difference of the two matched samples and the null hypothesis is that the distribution is symmetric about zero.

2.6.2. Correlation

When the participants perform a HRV condition relaxation we can assume the resonance attainment percentage with the photoplethysmograph data and calculate the Pearson correlation between this value and the three variables (thermo_diff, stai_diff, diff_all).

2.6.3. Time Dependency (Linear Regression)

Measurements were distributed over time, with one subject performing several relaxation exercises. We examined how the measured parameters in the two groups (HRV state condition and Free relaxation) changed as a function of the repeated performance of the exercise, grouped them accordingly, and then averaged the questionnaire difference scores. Linear regression was applied to detect trends. Most participants performed 2–4 measurements in total, and some completed more than 5. For linear regression, we used a weighting equal to the reciprocal of the standard deviation of the sample.

**3. Results**

Table 3 summarizes our results. The first and second columns show the results of the Wilcoxon tests, namely that both the HRV-BF and the free relaxation condition were able to reduce the stress level according to the STAIS-5 and DT, all $p < 0.015$.

In the third column, we see the result of the Mann-Whitney U test comparing the HRV-BF and free relaxation conditions. No significant difference was found between the two groups in stress scores (all $p > 0.0922$), indicating that both methods are similarly effective in reducing stress. In the last column are the results of the Pearson correlation. We can see negative correlations, meaning that the closer the participants could get to their resonance frequency, the smaller the relative difference in stress scores (indicating greater stress reduction). The highest correlation was with the STAIS-5 (−0.4175).

Figure 2 shows the fitted lines, with Figure 2A showing the comparison of the two methodologies on the thermo_diff variable, Figure 2B on the stai_diff variable and Figure 2C on the diff_all variable. The field around the points is the 'standard error bar' (standard deviation divided by the root of the number of elements). The relative differences of the STAIS-5 and DT slightly decreased over time in the HRV-BF group (i.e., the absolute values increased, so the stress reduction was greater), whereas in the free relaxation group the relative differences increased (i.e., they approached zero, so the absolute values decreased, which means that free relaxation could reduce stress less and less over time). The average value of the relative difference in stress scores (diff_all) also showed a decreasing trend in the HRV-BF group and an increase in the free relaxation group. Overall, this implies that the participants were able to learn the breathing technique better over time, so

that the HRV biofeedback condition contributes to better stress reduction, while the free relaxation becomes less and less effective in this regard.

**Table 3.** The results of our tests described with our three variables.

| Variable | Free Relaxation Difference (*p*-Values) | HRV Condition Difference (*p*-Values) | HRV Condition-Free Relaxation Difference (*p*-Values) | Progress Correlation (hrv) |
|---|---|---|---|---|
| thermo_diff | 0.0000 | 0.0000 | 0.4346 | −0.0577 |
| stai_diff | 0.0141 | 0.0001 | 0.0923 | −0.4175 |
| diff_all | 0.0001 | 0.0000 | 0.2630 | −0.3798 |

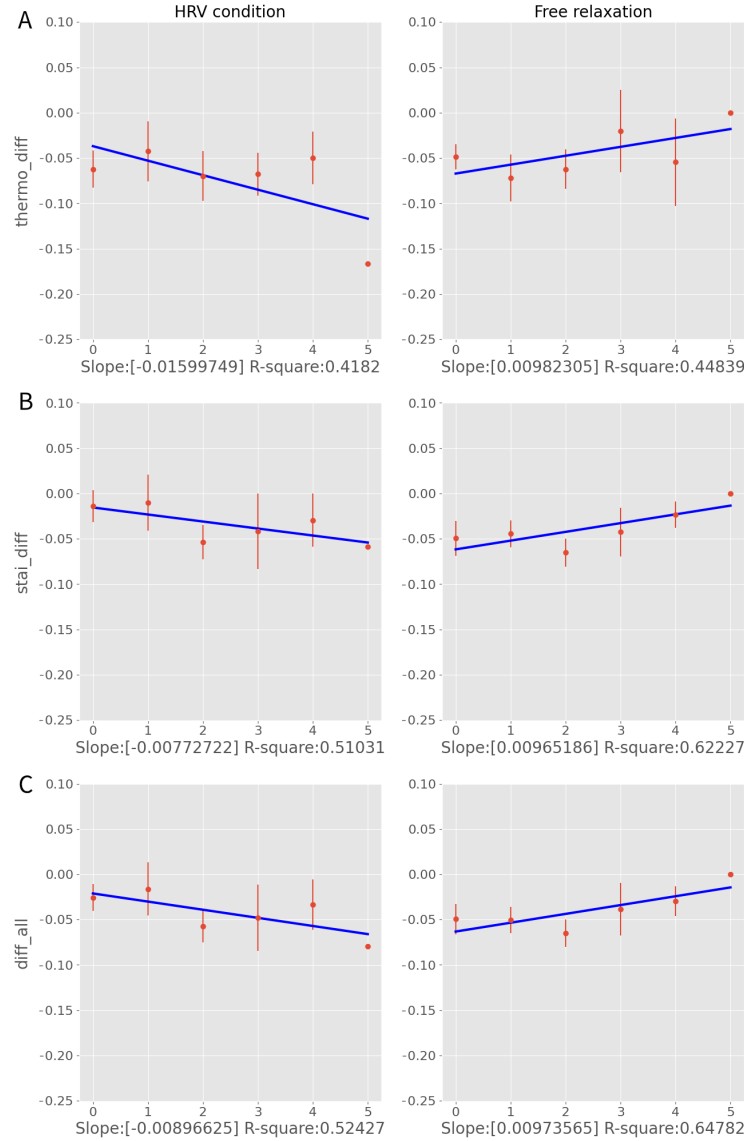

**Figure 2.** Linear regression fitted to variables. Each point is the average of the variables for each group. Each group was formed according to the number of times a given relaxation was performed. Subfigure (**A**) shows the comparison on thermo_diff, i.e., the relative difference in the Distress Thermometer (DT), (**B**) the comparison on stai_diff, the relative difference in the mean of the STAIS-5 discrate scale and (**C**) the comparison on diff_all, which describes the relative difference in the mean of the STAIS-5 scale and the Distress Thermometer (DT)

## 4. Discussion

There are some studies in the literature on the effectiveness of HRV biofeedback training. Most studies concluded that biofeedback training can reduce stress and anxiety symptoms generally as well as some other well-known methods, such as physical activity and mindfulness meditation [11] or autogenic training [8]. Our goal was to compare the HRV biofeedback method performed with our mobile application to the participants' usual relaxation methods, thus creating even more realistic working circumstances from the healthcare frontline to evaluate the efficacy of the method.

According to the participants' responses to the STAIS-5 and DT, both the HRV-BF and free relaxation methods were able to significantly reduce stress symptoms in the participants' workplace. There was no significant difference in effectiveness between the two methods when considering the average relative differences. According to Melumad and Pham [22], one explanation for this result may be that smartphone use can serve as a coping mechanism in stressful situations. In laboratory-induced stressful situations, subjects with higher stress levels were more likely to use their smartphones than subjects with lower stress levels, suggesting that smartphone usage provides psychological comfort that reduces stress [22].

We also examined correlation with time. The linear regression trends indicated that practicing HRV biofeedback might lead to better stress reduction over time. The reason for this change may be that during HRV biofeedback training, the subject implicitly learns the ability to reduce his or her stress level by controlling breathing. As implicitly learnt skills evolve over time, the ability to decrease stress by practicing abdominal breathing with pursed lips improves over time as well. All in all, our HRV-BF procedure can be distinguished from the lay methods workers typically use to relax themselves. Some studies have found efficacy after only one session [1,23,24], but usually multiple training and practicing sessions (ranging from a few to 50 or more) [1] are recorded to be effective. According to the original HRV biofeedback training protocol [5], training consists of weekly sessions and twenty minutes of home practice over five weeks. Furthermore, Lin and colleagues [8] found that even a single training session can lower stress. The results of our study suggest that HRV biofeedback training can be effective in reducing real work-related stress in frontline healthcare workers after a single training session. We also found that participants who were able to move their breathing closer to their own resonance frequency reported greater stress reduction. Similar result was found in a study [23], where the HRV biofeedback device generated scores based on the participants' pulse and breathing, the higher the score the closer the participants were to their resonance frequency. The study results showed that participants with higher scores had greater stress reduction. The time of the HRV biofeedback intervention can also be a relevant factor. The studies conducted by Prinsloo et al. [24] and Sherlin et al. [23] both found stress reduction after the intervention. In the case of the former study, the intervention lasted for 10 min, in the case of the latter, 15 min. Prinsloo et al. [24] assumed that the greater stress reduction observed in the other study could be explained by the longer intervention time. Our HRV biofeedback lasted 6 min. It might be interesting to conduct a study where the focus is on the effects of the intervention timing.

Another aim of ours was to disturb the measurements as little as possible, so no long training sessions were performed. Instead, the self-regulated stress-reducing activity of the workers became increasingly effective based solely on the operation of the algorithm. Theoretically, the personal training portion could be replaced with video instructions, making it even easier to use biofeedback stress reduction under everyday circumstances. Of course, further research is needed to support this theory.

One limitation of the study is that we did not consider demographic factors in the analysis, although the level of perceived stress could be influenced by sociodemographic factors, such as age, gender, marital status, work experience and educational level [25]. Studies indicate that healthcare professionals with work experience of 10 years or less perceivehigher levels of distress, than those who had over ten years of work experience [15,26].

Higher educational levels can serve as a protective factor against stress, as people with higher educational levels perceive lower levels of stress [27].

Another limitation is the sample size as it was less than the determined number with one person. In our opinion, the results are valid, but this should be taken into account when generalizing.

## 5. Conclusions

We have shown that we have succeeded in developing the methodology, designing and testing the algorithm, which has been shown to be suitable for stress reduction through HRV-based biofeedback training, and the application has been successfully tested under real-life conditions on various healthcare workers. The tests also proved our hypothesis that IT-only biofeedback-based relaxation sessions produce comparable stress reduction to spontaneous relaxation, even at the first use. However, real potential lies in the participant learning to consciously reduce stress with biofeedback from session to session, so that better results can be achieved than with spontaneous relaxation where such improvement has not been found. Of course, development and evaluation does not stop here. In the future, we plan to validate and refine the results obtained so far and further improve the effectiveness of the methodology.

**Author Contributions:** Conceptualization, J.D., V.B. and I.S.; methodology, J.D., V.B. and I.S.; software, Á.N.; validation, J.D., V.B. and I.S.; formal analysis, J.D., I.S., E.R. and E.A.H.; investigation, E.R., E.A.H. and Á.N.; resources, J.D., I.S., E.R., E.A.H., R.H., M.B. and A.L.; data curation, Á.N.; writing—original draft preparation, E.A.H., E.R., Á.N. and Z.S.; writing—review and editing, Z.S. and J.D.; visualization, Á.N.; supervision, J.D., V.B. and I.S.; project administration, I.S. and V.B.; funding acquisition, I.S. All authors have read and agreed to the published version of the manuscript.

**Funding:** Support by the European Union project RRF-2.3.1-21-2022-00004 within the framework of the Artificial Intelligence National Laboratory, the Ministry of Innovation and Technology NRDI Office within the framework of the Artificial Intelligence National Laboratory Program (RRF-2.3.1-21-2022-00004), by the Ministry of Innovation and Technology of Hungary from the National Research, Development and Innovation Fund, financed under the TKP2021-NVA funding scheme under project no. TKP2021-NVA-09.

**Institutional Review Board Statement:** The study was conducted according to the guidelines of the Declaration of Helsinki, and approved by the National Public Health Center, Department of Health Administration. Date: 28 March 2022; Registration Number: 6860-12/2022/EÜIG; Relevant Government Regulations: 23/2002, 235/2009. (X.20.), 531/2017. (XII. 29.).

**Informed Consent Statement:** Informed consent was obtained from all subjects involved in the study. Written informed consent has been obtained from the patients to publish this paper.

**Data Availability Statement:** The data presented in this study is not publicly available due to the nature of the consent signed by the participants.

**Acknowledgments:** Our team would like to thank András Bánhalmi and Bernát Gécs for their contributions in algorithm engineering and application development and Kata Németh-Rácz for her work in recruiting and managing the participants.

**Conflicts of Interest:** The authors declare no conflict of interest that influenced the content of this paper.

## Abbreviations

The following abbreviations are used in this manuscript:

| | |
|---|---|
| ICU | Intensive Care Unit |
| ED | Emergency Department |
| PU | Psychiatry Unit |
| PPG | Photoplethysmogram |
| DT | Distress Thermometer |
| HRV-BF | Heart rate variability biofeedback |

| AT | Autogenic Training |
| PA | Physical Activity) |
| MM | Mindfulness Meditation |
| PRESTINT | Predeployment Stress Inoculation Training |

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
