# Peer review of "Photoplethysmograph Based Biofeedback for Stress Reduction under Real-Life Conditions in Healthcare Frontline"

_applsci, doi:10.3390/app13020835_

Round 1

Reviewer 1 Report

I am really grateful to review this manuscript. In my opinion, this manuscript can be published once some revisions are done successfully. This study used 28 participants and popular statistical methods to compare the effectiveness of two interventions, i.e., free relaxation and heart rate variability biofeedback therapy. I would like to make three suggestions to improve the manuscript. Firstly, I would like to suggest the authors to improve the abstract by presenting specific information on the number of participants, the definitions and measures of interventions, the methods of comparing their effectiveness. Secondly, the sample size (28) is very small and this is a severe limitation of this study regarding external validation. I would like to suggest the authors to address this issue as a limitation of study in Discussion. Thirdly, the characteristics of participants (e.g., age, gender, education) can be important predictors for the level of stress and the effectiveness of interventions and the random forest can present the importance rankings of their major predictors. Indeed, the random forest is more accurate than conventional statistical methods. I would like to suggest the authors to address this issue as a limitation of study in Discussion. 

Author Response

Dear Reviewer,
Thank you for your comments. Based on them, we have made the following changes to the paper:
- the abstract has been expanded with more details on the number of participants and the specific methodology and evaluations
- in section 2.1 (Participants) and the Discussion we have added some new paragraphs explaining how we determined the number of the required subjects for the study and how this relates to the 28 people with whom we finally conducted the study
- in the Participants and Discussion sections, we have also included new paragraphs on the suggested demographic data and their relevance to the study results

Thank you again for the constructive suggestions, they helped us to improve our work.  

Reviewer 2 Report

The manuscript title “Photoplethysmograph based Biofeedback for Stress Reduction under Real-life Conditions in Healthcare Frontline” has scientific worth. Little modifications are required, especially in the methods and discussion section before acceptance of this manuscript.

Reviewer Comments:

1-      The abstract needs significant modification! From Line 6: the authors should explain the results/outcomes of their study, and write their major results and in the last 2-3 lines write down the conclusion of current study/ benefits of current study. Use short and meaningful sentences in the abstract section. 

2-      The headings 3 and 4 are part of methods section, so they should be changes as 2.3 and 2.4 respectively. Results will be heading 3 and discussion will be 4 and so on.

3-      In figure 2, I suggest authors to give subtitle to figure such as A, B, C; and explain the figure 2A, 2B, and 2C. Currently, the figure is very hard to read, the x-axis and y-axis figure title are also very small.

4-      In discussion section only 5-6 references were cited, I suggest authors to cite more studies to have a balanced discussion, that supports your results.

Author Response

Dear Reviewer,
Thank you for your comments. Based on them, we have made the following changes to the paper:
1. we have expanded the abstract with the proposed sections, the expansions detailing both the results and the scientific impact
2. we have implemented the proposed changes to the Heading.
3. the Figure mentioned has been rearranged with increased font size, its rows have been marked as A, B, C and the corresponding explanatory text has been expanded
4. in the Discussion, we have significantly increased the cited literature used, thus providing better support for our claims
Thank you again for the constructive comments, they helped us to improve our work.